# Monitoring Workloads of a Professional Female Futsal Team over a Season: A Case Study

**DOI:** 10.3390/sports8050069

**Published:** 2020-05-19

**Authors:** Carlos Lago-Fuentes, Alejandro Jiménez-Loaisa, Alexis Padrón-Cabo, Marián Fernández-Villarino, Marcos Mecías-Calvo, Bruno Travassos, Ezequiel Rey

**Affiliations:** 1Facultad de Ciencias de la Salud, Universidad Europea del Atlántico, 39011 Santander, Spain; carlos.lago@uneatlantico.es; 2Faculty of Sports and Education Sciences, Universidad de Vigo, 36005 Pontevedra, Spain; alexiscabo03@gmail.com (A.P.-C.); marianfv@uvigo.es (M.F.-V.); zequirey@uvigo.es (E.R.); 3Department of Sport Sciences, Sport Research Centre, University Miguel Hernández, 03202 Elche, Spain; alejandro.jimenezl@umh.es; 4Centro de Investigación y Tecnología Industrial de Cantabria (CITICAN), 39011 Santander, Spain; 5Departament of Sport Sciences, Universidade da Beira Interior, 6201-001 Covilhã, Portugal; bfrt@ubi.pt; 6Research Centre in Sports, Health and Human Development, CIDESD, 6201-001 Covilhã, Portugal; 7Portugal Football School, Portuguese Football Federation, 2784-214 Oeiras, Portugal

**Keywords:** team sports, women, session-RPE, performance

## Abstract

The aims of this study were to describe the external and internal workloads in a professional female futsal team during a whole season and to compare workloads during different periods of the season. Ten professional female futsal players (age 22.8 ± 4.3 years; 5.1 ± 2.4 years of experience; weight 61.9 ± 7.1 kg; height 1.66 ± 0.06 m) participated voluntarily in this study during the whole season. The internal workload was measured by the session-Rate of perceived exertion (session-RPE) method, while the external workload was indirectly measured by considering the training and match volume and the type of contents of each session over 43 weeks. Mean sRPE throughout the season was 319.9 ± 127.1 arbitrary units (AU). Higher internal loads (total weekly training load and strain) were reported during the pre-season compared with the in-season mesocycles (*p* < 0.05); meanwhile, the fifth to eighth mesocycles of the in-season showed an oscillatory pattern. Finally, Monday was the most-demanding session during the in-season period over the Thursday (*p* < 0.05; effect size: 1.33) followed by match day, meanwhile no statistical differences were reported during different sessions of the pre-season microcycle (*p* > 0.05). This study suggests that microcycles of pre-season present a stable load pattern, meanwhile workloads during the in-season period report a tapering strategy in a professional female futsal team.

## 1. Introduction

The popularity of futsal has been increasing around the world over the last fifteen years [1]. In fact, two new international tournaments have been recently approved by Fédération Internationale de Football Association (FIFA) and Union of European Football Associations (UEFA), as the Youth Olympic Games (both in male and female categories) and the European Female Futsal Cup to promote the development of this sport worldwide. Despite this global growth projection, only a few studies have been published in male futsal, and fewer in female futsal. Therefore, there is a need to investigate the training processes and the methods used by the technical staff to better understand the dynamics of futsal.

The training process is the best way for modulating athletes’ performance during the season [2]. Among other physical capabilities, it may help to increase strength, power, speed and endurance performance according to game demands. However, excessive amounts of training without sufficient recovery can be harmful for performance and injury risk [3]. In the same way, insufficient training can reduce the performance capabilities of one athlete or the whole team [2]. Thus, the development of monitoring and control strategies of sessions with an emphasis on the analysis of the workloads might be relevant for improving sports performance and to reduce the injury risk along the season. Usually, team sports workloads (both training and competition ones) are split into external and internal loads. The main external loads registered are: volume, distance covered, speed thresholds, accelerations and decelerations, while the most common internal loads are: oxygen uptake, heart rate, blood lactate, muscle load or ratings of perceived exertion (RPE) [2].

Several studies in futsal have analysed workloads using some of the aforementioned variables, such as heart rate, lactate, ventilatory threshold or heart rate variability in elite futsal players [4,5]. However, the most commonly used tool to measure training load in futsal is session-RPE (sRPE) [6,7,8,9], which has been shown to have acceptable reliability and internal consistency [10].

Analyses of the training load over a complete season in professional futsal players have shown large variations, particularly with a decrease of loads during the second half of the season [11,12]. Also, different patterns were registered during the pre-season in comparison with the in-season [8]. Likewise, other studies have found an inverse correlation among a lower RPE and higher performance levels in U20 professional futsal players [7]. However, these data exist only in male futsal players. Recently, Clemente and Nikolaidis [13] analysed the differences in training loads between sexes and sports (soccer and futsal), reporting higher intensity training loads in futsal than in soccer, but without differences among male and female futsal teams, although the female futsal team involved in that study competed in an amateur league and only four weeks of the in-season period were registered. With regard to this, it can be shown that several questions are needed to answer about the workload profile in professional female futsal teams. Taking into account the relevance of applying appropriate workloads to each sport modality and teams to improve physical performance and reduce injury risks, there is a need to analyse the workloads in a professional female futsal team due to the lack of scientific knowledge in this modality. In fact, to our knowledge, no previous studies have registered the workloads in a professional female futsal team during a whole season and compared the load profiles of a standard week of pre-season and in-season. Therefore, the aims of this study were: (i) to analyse and describe the external and internal workloads in a professional female futsal team during a whole season; (ii) to compare workloads during different periods of the season and, (iii) to describe and compare the load profile of a typical week during pre-season and in-season periods.

## 2. Materials and Methods

### 2.1. Study Design

This study followed a descriptive and prospective longitudinal approach over 43 weeks. Before starting the season, the project was presented to the technical staff of the team to be involved in the study, following by the players, who were informed of the study purposes and gave their informed consent. All research was carried out in accordance to the Declaration of Helsinki. The Investigational Review Committee of the Department of Physical Education and Sport Sciences of a Spanish university approved the research.

### 2.2. Participants

Thirteen professional female futsal players competing in the Spanish First Division Futsal League participated voluntarily in this study. Finally, three players were excluded because of injury or participating in less than 90% of sessions according to previous research [14], so ten players were fully registered (age 22.8 ± 4.3 years; 5.1 ± 2.4 years of experience; weight 61.9 ± 7.1 kg; height 1.66 ± 0.06 m; body mass index 22.3 ± 1.4 kg/m^2^). This team was playing in First Division for the first time, finishing 9th at the end of the season, and two players were called up to play for the Spanish National Team. Players had 5–6 weekly training sessions during the in-season with the team (7.2 ± 0.9 h/week). The team also played one official match per week during the in-season period.

### 2.3. Procedures

The external load was measured by the sum of training and match volume. The training volume was measured by the staff every training session by a chronometer, including the minutes of warm-up [13]. Player exposure during competition was also measured, taking into account the maximum of 40 minutes (official duration of futsal matches) [9]. Technical staff registered both data. The training contents were also registered, being classified according to previous scientific evidence in: strength training (exercises in the gym), power and acceleration (exercises done on the court with or without implements), specific endurance (exercises with the goal of improving aerobic and anaerobic power), preventive training (circuits designed to reduce injury risks), technical-tactical (main tasks with technical-tactical goals) and emotive-volitional tasks (exercises aimed at improving group cohesion and teamwork) [11]. Mesocycles (typical length of 5–6 weeks) were classified according to the specific criteria of technical staff in 3 criteria: increasing load (IL), maintenance load (ML) or decreasing load (DL) [15].

The internal load was monitored by the session-RPE (sRPE) method [16] thirty minutes after the end of each session reported on a 10-point RPE scale [5,9]. To reduce possible biases, caution was taken when collecting the scores, trying that the players did not listen to the scores of their teammates [17]. Care was taken to not modify the session routines in order to guarantee the ecological validity of the results [9]. sRPE was designed multiplying RPE by total volume in minutes (obtaining arbitrary units, AU) [10]. The monotony (daily mean split by standard deviation) and the strain (weekly load multiplied by monotony) were also calculated [17]. Total weekly training load (TWTL) was the sum of AU of each week’s session [12,14]. Finally, the TWTL of each mesocycle was obtained with the mean of the weeks’ TWTL during each mesocycle [7,12].

### 2.4. Statistical Analysis

We tested the assumption of normality using the Kolmogorov–Smirnov test. All variables were distributed normally, except for strain. Descriptive data is presented as mean (M) ± standard deviation (SD). Comparisons between daily workloads were analysed using analyses of variance (ANOVA), one for defining the differences among daily workloads during the pre-season and another ANOVA for the same during the in-season period. Differences between eight mesocycles were also calculated using ANOVA, followed by Bonferroni’s pairwise comparisons of the means. Effect sizes were calculated regarding differences between daily workloads, with >0.2, >0.5, and >1.2 being considered to represent small, moderate, and large differences, respectively [18]. All the data were analysed using IBM SPSS for Windows (version 20.0; SPSS Inc., Chicago, IL, USA). Statistical significance was set at *p* < 0.05. 

## 3. Results

The team played 17,566 minutes (including training sessions and matches) throughout the whole season with 480.2 ± 121.7 min per week, split into 157 training sessions, 30 matches of league and six of regional cup. The mean RPE throughout the season was 5.70 ± 0.64 and the sRPE, 319.9 ± 127.1AU. In addition, the TWTL was 2183.81 ± 838.45 AU, and the monotony and strain were around 1.00 ± 0.32 and 2419.75 ± 1961.25, respectively. Figure 1 presents the mean data about TWTL, monotony and strain throughout the 43 microcycles of the season.

Figure 2 illustrates the oscillatory pattern throughout the season by comparing the eight different mesocycles. A progressive decreasing load was observed throughout the season, showing significant difference in the TWTL among the first mesocycle and the remaining seven. Differences in the TWTL between the second and fourth mesocycle and the fifth, sixth and seventh were also shown (*p* < 0.05). The contents distribution is also highlighted in Table 1 showing that technical-tactical training volume increased during the eight mesocycles, while conditional training volume (strength, power and specific endurance) decreased along the season.

Figure 3 shows the mean weekly internal load presenting weekly periodisation of the pre-season and in-season periods. The weekly periodisation is different in both periods, showing a clearer weekly undulation in the in-season period. Pre-season training sessions are more similar, highlighting the match workload as the most demanding. The weekly periodisation on the in-season showed a higher AU on Saturdays with a decreasing load during the remaining days, showing higher loads on Monday and Tuesday over Thursday (*p* < 0.001; ES = 1.33 and *p* < 0.001; ES = 0.80, respectively).

## 4. Discussion

The aims of the present study were: (i) to analyse and describe internal and external workloads in a professional female futsal team during a whole season; (ii) to compare workloads during different periods of the season, and, (iii) to describe and compare the load profile of a typical week during pre-season and in-season periods. Based on this, the main findings of the present study were: (a) the technical-tactical training volume increased during the season, while the volume dedicated to conditional training decreased; (b) the internal load showed an oscillatory pattern in accordance with competitive periods; (c) the TWTL in mesocycle one was higher than the others followed by mesocycles two and four; (d) the workload showed a weekly periodisation with a tapering strategy during the in-season period.

The training contents showed a logical distribution in accordance with periodisation principles [2,8,19], where the conditional contents were more prominent in pre-season than in the other mesocycles. During the pre-season, Teixeira et al. [8] reported 50% of the time focused on technical-tactical contents, giving them more importance than in our study, and reporting double the time on power and strength. Conversely, technical-tactical tasks had more weight during the in-season mesocycles, similar to other studies, reaching over 60%–70% of training volume [11] or even up to 90% [20]. Strength training was one of the most important contents during the pre-season, with the sessions being reduced in volume during the subsequent mesocycles, as reported by previous authors [11,20]. However, as said above, other studies with professional male futsal players reported much more time dedicated to this capability during the pre-season [8], which could indicate the need to increase the volume of strength training in female futsal teams due to its relevance of sport performance. This shows some differences in contents distribution according to sex, with more relevance of strength training in men. Finally, the specific training content (technical-tactical and specific endurance tasks) seems to be the most important during the in-season period, similar to previous futsal studies [20,21].

The RPE is the most-used workload tool in futsal [6,7,8,9]. The mean RPE in our study was 5.70 ± 0.64 over the whole season. These data are very similar to previous studies in a season with youth male players (RPE = 5.5 ± 1.7) [20], a professional male team (RPE = 4–6) [21], or in different types of sessions with also professional male futsal players (physical session, RPE = 5.1 ± 0.7, technical-tactical session, RPE = 5.7 ± 0.8) [6]. However, our data are higher when compared to the study by De Freitas et al. [11] with professional male futsal players over 14 weeks (RPE = 3.3–4.3). Perhaps, with the exception of the latter, the mean RPE range 5–6 points could be considered as a reference value for the daily RPE over a season in elite futsal players.

The daily workload over a season was 320 ± 127 AU. Scott et al. [22] found similar values for daily average on Australian professional soccer players (297 ± 159 AU). However, Casamichana et al. [23] recorded higher values in semi-professional soccer players, with 462.4 ± 237.9 AU, similar to data recorded for professional futsal players 500 AU [6]. Notwithstanding, with regard to female futsal, our data are similar to previous research, which recorded an optimal training load among 350 and 450 AU [9]. According to this issue, it could be suggested that there are no differences on the daily workload among male and female elite team sports, perhaps slightly lower in female team sports. 

The mean TWTL was 2183.81 ± 838.45 AU throughout the season. Interestingly, our data match with previous studies with high level and professional male futsal players [24,25]. Nevertheless, these studies reported lower values than others with players of the same categories [7,14]. Both studies showed a TWTL among 2000–6000 AU and 3455–5243 AU, respectively. These differences could be due to the fact that these two studies had over 8 to 10 training sessions each week, compared to our study’s 5–6 sessions per week. Flat and Esco [25] recorded 2220–3000 AU in female soccer players with six training sessions per week, very similar to our findings. Thus, there seems to be a difference in total weekly workloads between sexes, likely due to the number of sessions. However, more studies are needed to validate our data as reference values for a female futsal team over a season. Finally, according to Foster [17], overtraining is more likely in weeks with workloads, monotony and strain over 4400, 2.2 and 6000, respectively. This study was performed with international competitive athletes but can be a reference of limits on workloads in sports training. In fact, similar values were only reached in our study during the second week, which highlights the relevance about managing workloads in order to reduce injury risks and optimise the conditional performance [2].

Regarding mesocycles, the workload decreased over the season, with higher loads during the first mesocycle. This matches with a study of professional female basketball players, preparing an international championship [26], and with one of the teams from the study of Teixeira et al. [8] in futsal. These increasing loads during the pre-season can accumulate higher amounts of fatigue in futsal players and, consequently, increase the injury risk [5,14] due to excessive loads combined with insufficient recovery [8]. For this reason, the pre-season should be accurately designed and controlled with different tools to ensure positive adaptations on futsal players [2] as well as reducing injury risks [8]. Moreover, the second and fourth mesocycles presented higher loads than the fifth, sixth and seventh ones. These results may be explained because of the sports programming theories [2,8,19], reaching higher loads on the first months, and the fourth mesocycle as a short pre-season during the in-season period [26]. However, another study found controversial results in professional female water polo players, with the last period of the season being the highest one, in comparison with the other three mesocycles [27]. This may be because the last phase of the season contained more intense and competitive matches, at both national and European levels. That is, the load control is fundamental to manage the athletes’ physical performance determined by the goals of the team and the structure and length of the season.

During the pre-season, the sRPE did not show significant differences among the days of the week, showing only differences between Saturdays (usually friendly matches) and Tuesdays. This can be explained by the greater variability and higher loads during the pre-season period, adjusting contents to balance training and recovery to optimise conditional performance [14,18,22]. During the in-season, the weekly workload distribution is shown in previous studies [17,28]. That is, the sRPE decreased during the week, with higher loads on Monday (first session of the week) after a full recovery day (Sunday), and the lighter loads on Thursday, to rest for Saturday’s match, as a tapering strategy, which attempts to ensure an adequate physiological response to competition [28].

Our study has some limitations as the number of external and internal load variables measured, as well as the number of players involved. More studies on professional female futsal players are needed, including more assessment tools to support these workloads, such as distance covered, the number of accelerations and decelerations, speed thresholds or changes of direction, among others. Likewise, further research should compare data from players in the First Division with lower divisions, given the differences shown by a previous study [29], as well as to include other important and specific variables in female athletes, such as the menstrual cycle and hormonal contraceptive use.

## 5. Conclusions

The present study shows the variation of training contents along the season, with the high relevance of technical-tactical tasks in a professional female futsal team with an inverse distribution of conditional volume. Likewise, workloads were higher during the pre-season mesocycles, reporting stable loads along the microcycles. During the in-season period, a tapering strategy was shown during the microcycle to achieve an optimal recovery for matches, reporting oscillatory patterns along the six mesocycles. Our findings may help to increase the knowledge about training periodisation strategies used in different contexts, such as professional female futsal teams.

### Practical Recommendations

As practical application for strength and conditioning coaches and for futsal staff in general, our study highlights the need to monitor workloads along the season and to analyse the weekly loads in the proper context (pre-season vs. in-season) in elite female futsal teams. Additionally, our data could serve as a reference of a microcycle type in this sport to compare the loads reached by a team during a season and to modulate/adjust their loads according to different goals. Finally, coaches and trainers are advised to apply appropriate workloads during the pre-season (especially reducing the volume), avoiding excessive weekly loads, which are related to greater injury risks and decrement of performance [5,14]. Also, staff should define a competitive microcycle with tapering strategies to ensure optimal physical performance in matches.

## Figures and Tables

**Figure 1 sports-08-00069-f001:**
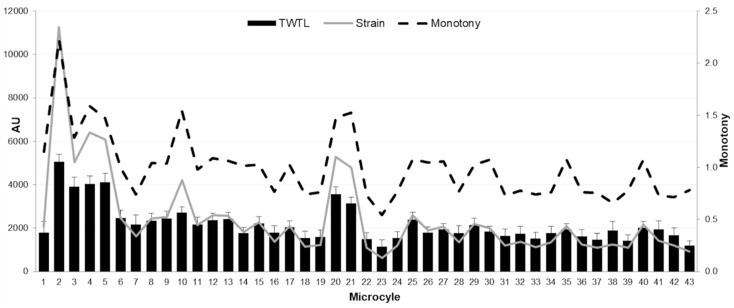
Description of the internal loads of female futsal players in each microcycle over a season. Notes: AU: Arbitrary units; TWTL: total weekly training load.

**Figure 2 sports-08-00069-f002:**
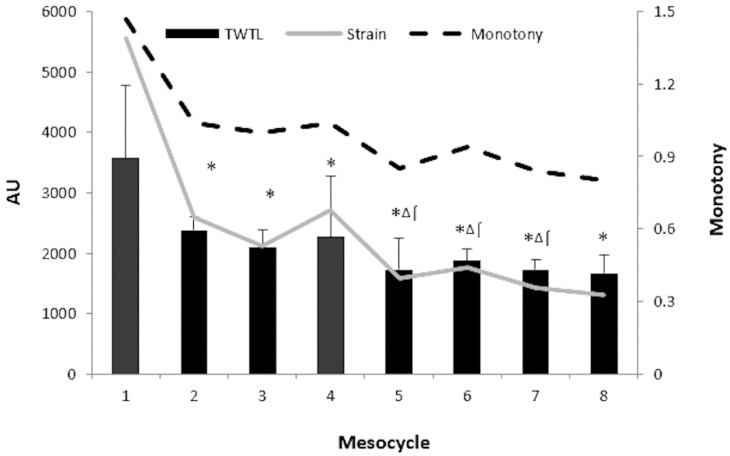
Differences in workloads parameters among mesocycles over the season. Notes: AU: arbitrary units; TWTL: total weekly training load; M1 and 4: preparatory period; M 2, 3, 5, 6, 7, 8: competitive period; * Significant differences between TWTL 1st mesocycle over the rest; ^⌠^ Significant differences between TWTL 2nd mesocycle over the rest; ^∆^ Significant differences between TWTL 4th mesocycle over the rest of mesocycles.

**Figure 3 sports-08-00069-f003:**
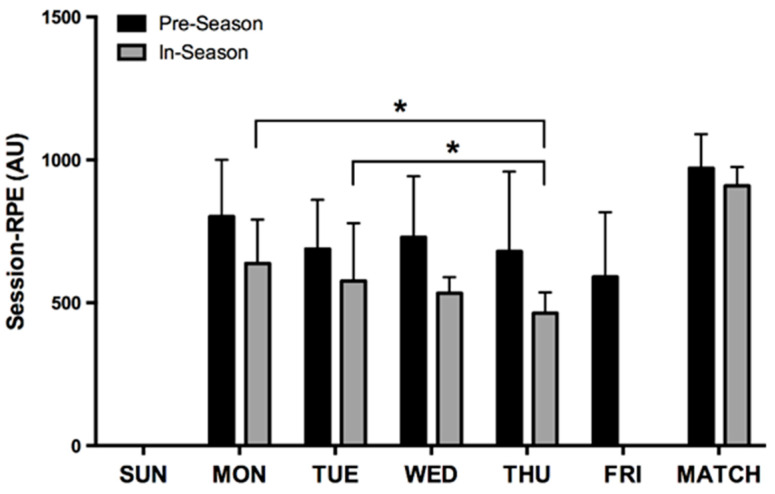
Daily workload during pre-season and in-season. * Significant differences.

**Table 1 sports-08-00069-t001:** Description of general characteristics in each mesocycle over a season.

Mesocycle	1	2	3	4	5	6	7	8
Type	IL	ML	DL	IL	DL	IL	ML	ML
Weeks	6	5	7	5	4	5	5	6
Matches	7	5	6	1	4	5	5	6
Period	PP	CP	CP	PP	CP	CP	CP	CP
Strength	315	60	70	22	20	30	10	0
Power	95	105	100	85	85	75	70	60
Specific Endurance	585	145	270	290	80	210	160	125
Preventive	490	480	415	235	245	260	205	305
Technical-Tactical	1620	915	1165	775	860	935	815	1090
Emotive-volitional	655	430	515	200	225	275	350	270

**Notes:** Contents are represented in minutes. IL: increasing load; ML: maintenance load; DL: decreasing load; TWTL: total weekly training load (in Arbitrary Units); PP: preparatory period; CP: competitive period.

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
