# Peer review of "Monitoring Workloads of a Professional Female Futsal Team over a Season: A Case Study"

_sports, 2020, doi:10.3390/sports8050069_

Round 1
Reviewer 1 Report
The authors sought to investigate the changes in external and internal training loads in a group of professional futsal players across a 43-week season. The data presented could be very useful for strength and conditioning practitioners. However, in its current form the manuscript does not provide sufficient detail/information to allow the reader to fully understand what was done (i.e. how many of the dependent variables were calculated) or even why certain analyses were performed (i.e. unclear rationales for the statistical assessments). Therefore, I am requesting that the authors revise the manuscript relative to the following issues:
Title: Correct the spelling errors in the title.
Abstract, line 21: Does 5.70 refer to the mean RPE using the CR-10 scale & 319.9 is the session RPE? Please be specific.
Abstract, lines 23-25: Please be more specific here. Are the correlations between the internal and external training loads? It reads as though you have merely correlated training loads (interval scale) with the micro- and mesocycles (ordinal scale).
Abstract, line 27: Change to “…low cost tool to monitor training loads…”
Introduction, line 35: Change to “…female futsal. There is therefore a need to investigate…”
Introduction, line 37: This sentence needs to be rewritten to allow the appropriate meaning to be conveyed.
Introduction, line 39: Change to “…training without sufficient recovery can be harmful…”
Introduction, line 42: Change to “…sessions with an emphasis on the…”
Introduction, line 44: Change to “…sports, are divided into external and internal…”
Introduction, line 48: Change to “…workloads using some of the aforementioned variables, such as…”
Introduction, line 50: Change to “…the most commonly used tool to measure training load in futsal is session-RPE (sRPE) [6-9], which has been shown to have acceptable reliability and internal consistency [10].”
Introduction, line52: Consider changing to “Analyses of the training load over a complete season in professional futsal players have revealed large variations, particularly during the second half of the season [11, 12].” But you need to specify whether these are assessments of external or internal training loads.
Introduction, lines 56-57: You need to support this statement with an appropriate citation.
Introduction, lines 61-63: Change to “…players during a whole season and compared the load profiles from pre-season to those recorded in-season. Therefore, the aims of the study were…”
Materials and Methods, lines 69-72: Does the anthropometric and demographic data presented here apply to the original 13 players or the final 10? Please only report this data for the final 10 players.
Materials and Methods, line 72: Change “…assisting…” to “…participating in…”
Materials and Methods, lines 80-81: You need to provide much greater detail on the methods used to record training volume. This is a key dependent variable in your study design and you therefore need to ensure that you provide sufficient details to allow replication of your methods by future researchers.
Materials and Methods, lines 82-83: Again, provide information as to the methods used to categorize the training sessions. It is not enough simply to present a citation that may contain the information.
Materials and Methods, lines 86-91: It appears that you have recorded both the RPE and session-RPE. What is the rationale for that? It is not presented as one of the aims of the study, so why is it necessary to present these separately. Do you think that they will provide different information? If so, please enlighten the reader. You also need to expand upon the methods used to calculate both “monotony” and “strain” here as these are your dependent variables.
Materials and Methods, line 89: Change to “…not to modify the session routines in order to guarantee…”
Materials and Methods, lines 98-99: You need to provide a better description of the ANOVA model here i.e. was it two one-way ANOVAs, one for the pre-season and one for the post-season, or was it a two-way ANOVA (and if so, how levels on each factor?)?
Materials and Methods, lines 101-103: Here you state that correlations between RPE and training volume as well as between RPE and TWTL, but there is no mention of session-RPE. Are we to assume that “RPE” is in fact sRPE and you are therefore not collecting RPE and sRPE data?
Materials and Methods, line 101-103: If I am to assume that RPE is sRPE, what is the purpose of exploring the relationship between sRPE and TWTL, given that the calculation of TWTL is based upon sRPE? Even if RPE is actually RPE, why are you interested in the strength of its relationship with TWTL? What are you doing to RPE (sRPE?) to allow you to perform this correlation with TWTL? Aren’t the values the same (presumably you are averaging the RPE values over a given duration)? Such an analysis does not appear to align with the aims of the study presented at the end of the Introduction. You need to provide a much more detailed explanation of how the data were treated and why the analyses were performed in order to help the reader.
Results, line 110: Here the reader is introduced to the variable of “monotony” for the first time. Any dependent variable needs to be appropriately presented in the Materials and Methods section.
Results, Tables 1 & 2: I think that you would better serve the reader by presenting this data in graph form rather than table form.
Reviewer 2 Report
Monitoring workloads of a profesional female futsal
tea mover a season: a case study
I guess the title should be “Monitoring workloads of a professional female futsal team over a season: a case study”.
Your study is of interest and gives interesting input to the field of applied physiology and training control. With 13 subjects, this is not a representative study, but non the less gives important information about futsal training load.
You cite Borg (1982) when describing mesocycles (Line 85), that reference is not about describing training load per. se. I suggest You find another reference, or makes the connection with the Borg reference clearer.
The variables “strain” and “monotony” is poorly described and should be more explained in the text. I would also suggest using the original reference, of Foster (1998).
For normality, Kolmogorov-Smirnov test was used, with Lillefors correction? And how did You treat the strain variable as it was not normally distributed? SPSS from v. 19 is cited as IBM SPSS, Armonk, NY, USA. Do the journal use italics or not for p?
Your presentation of results is clear, but maybe the data in Table 2 would profit of being presented graphically, at least some of the variables? This would show the oscillations through the mesocycles more easily.
Figure 1 needs more explanations in the figure text, what do the * stand for, what p-level do they represent?
“The training content showed a logical distribution in accordance with periodization principles” (Line 157) should be supported be some references.
I do have some minor comments in the text, please look at the attached file.

Reviewer 3 Report
Monitoring workloads of a professional female futsal team over a season: a case study
Title
Contains two spelling errors – professional and team over. Please perform a full check of the manuscript
Abstract
L22 both internal and external loads were higher? What were the other mesocycles? Please provide an example or some detail
L25 ‘meaningful correlation’ does not make sense
L26/27 the aim of the study does not indicate you’re are trying to evaluate the meaningfulness of difference measures. Its purely description, so you have to alter your conclusion
Introduction
L31 meaning of ‘relevance’ is unclear. ‘Last years’ – please be specific
L37 ‘best key’ is unclear
L52 this paragraph lacks detail. You’re just stating studies without giving the reader information regarding where your proposed study sits within the literature
L53 ‘oscillatory pattern’ defined in parenthesis for clarity
L57 “reporting high intensity” – this doesn’t make sense. I’m going to stop commenting on the elements of this that don’t make sense as there are far too many. Please read over your manuscript carefully.
Methods
L80 this first paragraph should go into a study design section
L84 typical mesocycle length?
L86 the session RPE method?
L91 include calculations for strain and monotony
Results
Some of the data in Table 2 would be better displayed in a graph with means and SDs. Also, in the in text data states significance and non-significance but provides no P values.
Figure 1 – might it be useful to include effect sizes on this data?
Figure 2 – state which panels A b and C are in the figure label. Also the figure text isn’t legible
Discussion
L148 – change goals to aims
L167 – given the data is in females, you need to state how this different for your study i.e. what the focus was for your athletes
L169 – you didn’t have a sample of males, so you can’t compare between sexes
L184 – why might this be different compared to the other samples, in your study?
L194 – in what sample and sport was this data for? Surely there is a large variability in this
L230 – you can comment on the usefulness of RPE. It seems to be outside of the scope of your data
Please include practical recommendations in your conclusion
Round 2
Reviewer 1 Report
I still have a number of issues that I would like the authors to address. In particular, I think that the authors need to justify their measure of external training load (duration of the training sessions). While the method of determining the different training sessions is highlighted by the authors, the duration only provides a limited indication of volume; in the absence of any measure of intensity during the sessions this metric is largely meaningless. Indeed, the authors note in their Introduction that typical measures of external training load include distance covered, speed thresholds, acceleration and deceleration activities; simple workout duration is not typically used as an assessment of external training load. This limits the relevance of the study.
Introduction, line 54: Change to “…decelerations, while the most common…”
Introduction, line 65: Change to “…Likewise, other researchers have found a correlation…”
Introduction, line 66: Change to “…but all of these previous studies were conducted with male…”
Introduction, lines 67-74: The two sentences presented here are largely the same. Re move one of them in your revision.
Materials and Methods, line 86: Change to “…involved in the study…”
Materials and Methods, 87: Change to “…study, following which the players were informed…”
Materials and Methods, line 110: Change to “…(exercises in the gym)…(exercises done on the court…”
Materials and Methods, line 114: You are measuring external training load simply as the time spent engaging in the specific activities of strength training, power and acceleration, specific endurance, etc. Yet you note in line 53 of the Introduction that the most common measures of external loads are distance covered, speed thresholds, acceleration and deceleration. You have note employed these methods. Does this invalidate your findings, certainly when it comes to assessments of external training load (simply assessing the duration of the workouts does not provide a useful assessment of the external training load).
Results, line 142: When you state “played 17,566 minutes through the whole season”, does this refer to the competitive game time or the cumulative duration of the training sessions in its entirety? This needs to be made clear.
Results, Figure 1: The axis for the strain scores are not identified.
Results, lines 157-162: You note that the results of the internal training loads follow an oscillatory pattern and also note that “increasing importance of technical-tactical training to the detriment of conditional training volume” (Table 1), but you have not presented the results of your ANOVA model. These are substantial omissions and needs to be added because your interpretation cannot be considered entirely objective.
Results, Figure 2: It would be useful to identify the demarcation between the pre-season and in-season mesocycles.
Results, Table 1: You need to add the units of measurement for the training sessions (presumably, minutes).
Reviewer 3 Report
First of all, it is very hard to know where the changes have been made because this document is in track changes and the authors have not put line numbers in their responses. I suggest you amend this in future.
I’m still not really sure what benefit the study has. For example, its obvious that training loads will vary across the season. You need to make a better case as to why this is important and where you study expands on previous work. There are some useful recommendations in the discussion which you could allude to in the introduction.
L25-26 indicate that P < 0.05 or > 0.05
L30-31 again, the conclusion to this study is obvious. That doesn’t mean the study doesn’t need to be done, but you need to make more of your data
L62 can you indicate the direction and reason of the variations
L66-67 “but all these researches are conducted with male futsal players”. Change to “ however, this data exists only in male futsal players”
L71-72 this reads almost identically to the previous sentence. Can you also expand on why there might be differences between males and females
L110 should be exercises in the gym. This paper still needs a fully check. Please review
Its really hard to tell what is happening within the figures because this is in track changes
Why do the authors not think effect sizes are needed?
L212 please clarify “need to grow up the relevance”
L287 change ‘relevance’
L291 clarify what you mean by dose better workloads
L292 change bigger to greater. Also, I’m not sure this is a recommendation from your work as you’ve not provided the injury type data
Round 3
Reviewer 1 Report
The authors have improved the manuscript with their revisions.
I only have a very small number of corrections:
Line 91: Remove the period (full stop) from the end of this line.
Line 177: Change to "...with the sessions being reduced in volume during the subsequent mesocycles, as reported by previous authors."
Line 183: Do you mean "in-season" as opposed to "season" here? And when you say the 'specific training content" do you mean strength training? This sentence is confusing.
Line 199: Change to "...perhaps slightly lower in female..."
Line 233: Change to "...shown in previous studies."
Line 237: Remove "However" from the beginning of the sentence.
Reviewer 3 Report
All amendments made.
